# Bayesian Object Models for Robotic Interaction with Differentiable Probabilistic Programming

Krishna Murthy Jatavallabhula[*‡], Miles Macklin[1], Dieter Fox[1], Animesh Garg[1], and Fabio Ramos[1]

[1]NVIDIA

**Abstract:** A hallmark of human intelligence is the ability to build rich mental models of previously unseen objects from very few interactions. To achieve true, continuous autonomy, robots too must possess this ability. Importantly, to integrate with the probabilistic robotics software stack, such models must encapsulate the uncertainty (resulting from noisy dynamics and observation models) in a prescriptive manner. We present *Bayesian Object Models* (BOMs): generative (probabilistic) models that encode *both* the structural and kinodynamic attributes of an object. BOMs are implemented in the form of a differentiable probabilistic program that models latent scene structure, object dynamics, and observation models. This allows for efficient and automated Bayesian inference – samples (object trajectories) drawn from the BOM are compared with a small set of real-world observations and used to compute a likelihood function. Our model comprises a differentiable tree structure sampler and a differentiable physics engine, enabling gradient computation through this likelihood function. This enables gradient-based Bayesian inference to efficiently update the distributional parameters of our model. BOMs outperform several recent approaches, including differentiable physics-based, gradient-free, and neural inference schemes. (**Webpage**)

**Keywords:** Simulation, Probabilistic programming, Differentiable physics

## 1 Introduction

When presented with a novel object, humans are quickly able to discover a number of structural properties and functional attributes. As children, they play around with objects, garnering evidence from each interaction to gradually build powerful mental representations of objects, their constituent parts, and the physical constraints imposed by their configuration. Most robots, however, rely on pre-specified kinodynamic object models for interaction planning.

We propose an interactive mechanism by which a autonomous agent can build "Bayesian object models" (BOMs)—probabilistic representations of scene/object structure and dynamics—using an *analysis-by-synthesis* framework [1]. In our study, robots are equipped with rich world models (forward predictive models of kinodynamics) in the form of *differentiable probabilistic programs*. We formulate the construction of BOMs as a Bayesian inference problem over this probabilistic program (*generative model*), given a small number of real-world interactions (*evidence*).

A *probabilistic program* is a program written in a special purpose language—a probabilistic programming language (*PPL*)—that augments general-purpose programming languages by providing explicit constructs for sampling and conditioning over random variables. The primary goal of a PPL is not just to provide a flexible specification mechanism for probabilistic models; but also to automate Bayesian inference. An increasing number of modern PPLs [2–5] natively support automatic differentiation, due to the widespread adoption of differentiable computing in machine learning applications [6–8]. We refer to programs written in such a PPL—that leverage gradient-based probabilistic inference—as differentiable probabilistic programs.

---

[*]Work done as part of an internship at NVIDIA.

6th Conference on Robot Learning (CoRL 2022), Auckland, New Zealand.

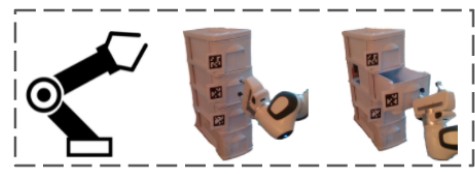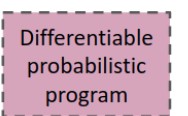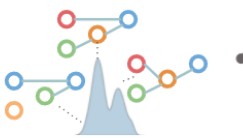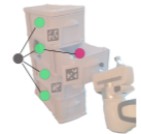

(a) Robot interacts with objects in the scene    (b) Observes object/part trajectories

(c) Infer **Bayesian World Model** (BWM) to best explains the observations by learning a probability distribution over scene structure (tree) and physical properties

Figure 1: **Bayesian Object Models** (BOMs): Robots interacting with the world observe the dynamics of previously unseen objects. We encode an explicit understanding of the world (a *generative model*) in the form of a *probabilistic program* that models latent scene structure, dynamics, and observation models. Probabilistic programming enables us to automate several aspects of Bayesian inference, resulting in the construction of BOMs from a small number of robot-environment interactions (8-20 actions in our experiments).

Our generative model is one such program, relating latent scene structure (often a tree) and kinodynamic attributes governing object states. The state-space trajectories generated by the model, in conjunction with a small set of trajectories obtained by interacting with the scene, are used to formulate an observation likelihood for Bayesian inference. Importantly, this likelihood is differentiable, allowing for efficient gradient-based probabilistic inference schemes. This is in contrast to the set of approaches recently beind unified under the term *simulation-based inference* - SBI, where the generative model includes a *black-box* simulator.

While object 'dynamics' can be differentiated through (enabled by the emerging volume of work on differentiable physics simulation [9–18]), these methods cannot compute gradients with respect to the object's 'structure'. We overcome this by devising a differentiable procedure for sampling tree-structured random variables. Our generative model includes the following latent variables: graph structure, edge types, object physical properties. Our framework is the first to learn a distribution over *both* the scene structure and its physical parameters – essential for interoperation with current robot software stacks that rely heavily on probabilistic modeling and inference.

## 2 Related work

Many approaches have tackled interaction planning for **objects with unknown articulations**. They typically only capture point estimates of articulation parameters [19–28], or probabilistic models of articulation kinematics [29–31]. Recent approaches leverage differentiable simulation for recovering articulation parameters [32–35, 16]. Our BOMs capture distributions over tree structures of objects, and their kinodynamic parameters via efficient gradient-based probabilistic inference.

Our work is related to the **causal inference and structure discovery** literature [36–44], specifically building on the characterization of directed acyclic graphs (DAG) presented in [39–41], extending these ideas to differentiably sample tree-structures. Other approaches to causal structure discovery include neural inference [37, 38, 42–44] and symbolic regression [45–47]. We rely on an explicit, differentiable parameterization and demonstrate its efficacy in robotics problem domains.

**Probabilistic programming languages (PPLs)** have long been used in the Bayesian inference community for model specification and inference [48–50]. Extensive efforts into modern differentiable programming frameworks [2–5] has also resulted in the development of modern PPLs with native support for autodifferentiation. We use the Pyro PPL [2] due to its interoperability with Pytorch [6].

Modern PPLs have yet to significantly impact **problem domains in robotics**. 3DP3 [51] proposes a generative model of depth images to recover distributions over object poses; DurableVS [52] enables unsupervised visual servoing; and Mirchev *et al.* [53] enable 3D SLAM.

**Simulation-based inference (SBI)** is a recently-coined term [54] that attempts to unify multiple lines of work leveraging simulation models for probabilisitic inference. Originally referred to as likelihood-free inference, SBI approaches have thus far assumed black-box simulators. In this work, however, we propose a new approach to SBI, leveraging white-box simulators interfaced with differentiable probabilistic programs.

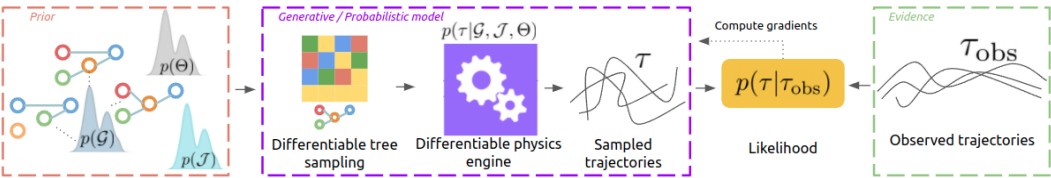

Figure 2: **Differentiable probabilistic programming for learning BOMs**: A probabilistic program (shown above) samples random variables encoding a scene structure (a graph $\mathcal{G}$), physical constraints $\mathcal{J}$, and object physical properties $\Theta$. These sampled variables are used to initialize a differentiable physics engine that simulates the dynamics of the environment and computes a likelihood function. A key idea here involves a differentiable sampling procedure for the graph-structured variables, which allows us to perform gradient-based (variational) inference of the posterior over the random variables of interest.

## 3    Bayesian Object Models

**Problem definition**: A robot interacting with an object (containing multiple moving parts) applies control actions $\mathbf{u}$ and records part trajectories $\tau_{\mathrm{obs}}$ (here, point cloud streams). We represent the scene as a graph (specifically, a tree) $\mathcal{G} = (\mathcal{V}, \mathcal{E})$, as is the norm in simulation engines for robotics. Each node $v \in \mathcal{V}$ in the graph denotes an object part (if $v$ is a leaf node), an object part collection or an entire object (if $v$ an internal node). Each edge $e \in \mathcal{E}$ in the graph denotes a joint-type $j_e \in \{rigid, prismatic, revolute\}$ with parameters $\theta_e$. Let $\Theta$ denote the set of all physical parameters over all edges in the graph, and $\mathcal{J}$ denote an enumeration of joint types, i.e., $\Theta = \{\theta_1, \cdots, \theta_{|E|}\}$, $\mathcal{J} = \{j_1, j_2, \cdots, j_{|E|}\}$. We aim to recover a distribution over scene structures, and over the physical parameters of the structure $\Theta$ (object relationships, joint types, joint parameters) that best explain the observations $\mathbf{z}$, i.e., $p(\mathcal{G}, \mathcal{J}, \Theta | \tau_{\mathrm{obs}})$.

$$p(\mathcal{G}, \mathcal{J}, \Theta | \tau_{\mathrm{obs}}) = \eta\, p(\tau | \mathcal{G}, \mathcal{J}, \Theta)\, p(\mathcal{G})\, p(\mathcal{J})\, p(\Theta) \qquad (\eta \text{ is a normalizing constant}) \qquad (1)$$

**Bayesian Object Models (BOMs)** are generative (probabilistic) models representing the distribution $g_\phi(\tau | \mathcal{G}, \mathcal{J}, \Theta)$ over object structure and kinodynamic parameters. Specifically, a BOM is a differentiable probabilistic program $g_\phi$ parameterized by $\phi$. Samples $\tau$ drawn from this model represent object part trajectories, and are used to compute *likelihood* functions $p_\phi(\tau | \mathcal{G}, \mathcal{J}, \Theta)$ (differentiable w.r.t. $\phi$). They can therefore be employed in gradient-based probabilistic inference schemes to infer $\phi$, conditioned on real-world observations $\tau_{\mathrm{obs}}$.

### 3.1    Likelihood function

An optimal choice of model parameters $\phi$ for our BOM should *best explain* the observed trajectories $\tau_{\mathrm{obs}}$ (*evidence*). Hence, a straightforward choice of likelihood function would be one where the maximum likelihood estimate amounts to minimizing a mean-squared error between sampled trajectories (from the model) and observed trajectories. We therefore assume that our observation errors are Gaussian distributed with inverse variances $\beta_*$, which results in the following likelihood

$$p(\tau_{\mathrm{obs}} | \tau) = p(\tau_{\mathrm{obs}} | \mathcal{G}, \mathcal{J}, \Theta) = \prod_{t=0}^{T} \exp\left(-\beta_t \|\tau_i - \tau_{\mathrm{obs},i}\|^2\right) \text{ where } \tau_i \sim g_\phi(\tau | \mathcal{G}, \mathcal{J}, \Theta) \qquad (2)$$

Gradient-free Bayesian inference in the parameter space $\phi$ of this model is computationally expensive and performs poorly in practice (see Sec. 4). The tree $\mathcal{G}$ involves several discrete-structured random variables. Further, each edge in $\mathcal{G}$ has several properties (joint type, friction, damping, axis parameters), resulting in high sample complexities. BOMs alleviate this complexity by presenting a differentiable likelihood computation scheme, including a differentiable greedy mechanism for sampling tree-structured random variables, a differentiable physics engine to model time-evolution of object state, and observation models.

### 3.2    Choice of prior distributions

We assume the following choices of prior distributions within our probabilistic program. These families of prior distributions are general and are only used as scaffolding for our inference engines. One may also explore other specialized choices in return for further performance improvements.

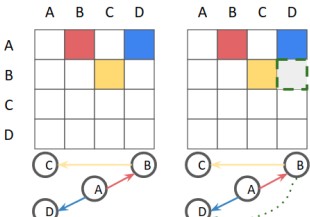

Figure 3: Our **differentiable tree sampling** procedure builds on the differentiable DAG sampler proposed in [41] and imposes additional constraints to enforce tree structure. (*Left*): A tree and its upper triangular adjacency mat (here, $P = I$. (*Right*): Adding the dashed edge makes the tree a DAG. Our sampling procedure performs an opportunistic rounding step to prevent edges like this one

**Prior over tree structure**: We assume that all scene structures (trees) are equally likely. This results in a uniform distribution over the set of $(N + 1)^{(N-1)}$ trees with directed edges rooted at a given node (Cayley's theorem). The prior probability of any sampled tree with $N$ nodes (excluding the root) is thus, $p(\mathcal{G} = G) = \frac{1}{(N+1)^{(N-1)}}$.

**Prior over joint types**: We parameterize joint types using a categorical distribution $p(\mathcal{J}) = \mathrm{Cat}\{\texttt{rigid}, \texttt{prismatic}, \texttt{revolute}\}$. While gradient-based optimization through a categorical distribution techically requires *enumeration* (i.e., duplicating the computation graph for all possible choices of $\mathcal{J}$), we find that using a softmax function and a straight through estimator instead works well in practice.

**Prior over kinodynamic parameters**: Each edge has continuous physical parameters indicating its friction, damping, and axis parameters (a 6D *screw* vector). Each node has a center-of-mass parameter. These parameters over the entire graph are lumped into a vector $\Theta$, which is assumed to be a standard normal distribution $\Theta \sim \mathcal{N}(\mathbf{0}, \mathbf{I})$. We also attempted using a uniform distribution and observed that this leads to slightly inferior performance.

### 3.3 Generative Model

Our generative model $g_\phi(\tau | \mathcal{G}, \mathcal{J}, \Theta)$ relates the observed trajectories $\tau_{\mathrm{obs}}$ to latent scene structure $\mathcal{G}$ and kinodynamic parameters $\mathcal{J}, \Theta$. We implement all modeling aspects in the Pyro PPL [2, 4].

**Overview**: Our generative model samples object part trajectories as follows. First, we independently sample a graph structure $\mathcal{G}$ and kinodynamic parameters $\mathcal{J}, \Theta$ from the prior distributions $P(\mathcal{G}), P(\mathcal{J})$, and $P(\Theta)$ respectively. Next, we evolve the object state using a physics engine $\Psi$ given external input forces. The physics engine produces object states, from which we sample a trajectory sequence $\tau$ by adding white Gaussian observation noise.

The key challenge in constructing this generative model is to ensure that the output trajectories are all differentiable with respect to the parameters $\phi$ of $g_\phi$. We do this by (a) proposing a differentiable mechanism to sample tree-structured variables, and (b) leveraging advances in differentiable physics simulation to compute gradients through the physics engine $\Psi$.

**Efficient differentiable sampling of tree-structured variables**: Recently, multiple schemes have been proposed to efficiently sample DAG-structured random variables [39–41]. All of these approaches rely on the fact that a graph $\mathcal{G}$ is a DAG if and only if its adjacency matrix $A$ is nilpotent [55][2]. This means $A$ can be factorized into a permutation matrix $P$ and an upper-triangular matrix $U$, such that $A = PUP^T$ [39, 41].

Our differentiable tree sampler is based on the differentiable DAG sampler DP-DAG [41], extending it using a greedy projection step to cast the DAG into a tree. We first sample a directed acyclic graph (similar to DP-DAG [41]) and present a greedy projection step to obtain a tree.

*1. Differentiable DAG sampling*: Each element of $U$ is sampled using the Gumbel-Softmax trick; $u_{ij}$ ($j \geq 1, 1 \leq j \leq i$) are reparameterized as

$$u_{ij} = \frac{\exp\left(\frac{\gamma_{ij} + \log(\pi_{ij})}{\tau}\right)}{\sum_k \exp\left(\frac{\gamma_k + \log(\pi_k)}{\tau}\right)}$$

---

[2]A square matrix $A$ is nilpotent if $A^n = 0$ for a positive integer $n$. In terms of graphs this means that powers of adjacency matrices (which indicate $n$-hop neighbours) do not result in cycles.

where $\gamma_*$ are drawn from the Gumbel distribution $\gamma_* \sim \text{Gumbel}(0, 1)$. Autodifferentiation through this is enabled by the straight-through estimator [56, 57]. While the above approach results in a (soft) adjacency matrix, a hard thresholding typically results in a DAG.

*2. Opportunistic rounding*: To reduce a DAG to a tree, one needs to impose additional constraints; specifically that a unique path exists to each node in the DAG from the root node. The entries $u_{ij} \in U$ must be structured such that, all entries $u_{kj}$ $(k > i)$ are set to zero, if $u_{ij}$ is 1. We achieve this by an opportunistic rounding scheme; for each column $j$, we find the first entry $u_l j$ in the soft upper-triangular matrix $U$ above a threshold $\delta$ and round it up to 1, while simultaneously rounding all entries $u_{mj}$ $(m < l)$ to zeros. If no entry in column $j$ exceeds $\delta$, we pick the first non-zero value in the column to round up to 1. Gradients through the rounding step are propagated, again, using the pathwise derivative (straight-through estimator). This scheme provably samples a tree.

**Differentiable physics engine**: Given a graph structure $\mathcal{G}$ and kinodynamic parameters $\mathcal{J}, \Theta$, a differentiable physics $\Psi$ engine simulates the time-evolution of object states. Our discrete-time physics engine simulates phenomena such as articulations, mass-spring systems, electrodynamics, and produces a trajectory $\tau$. We employ a deterministic dynamics model – all of the stochasticity in our generative model arises due to the latent scene structure and kinodynamics. To account for any unmodeled dynamics effects, we add an observation noise term to each state in the trajectory. We find that such a stochastic observation model works across simulated environments, and also in real-world scenes. A distinct aspect of our approach (compared to prior work in differentiable physics simulation) is that our differentiable physics engine is embedded within a PPL. This allows the additional flexibility of sampling from the generative model, conditioning on observed values of random variables, and also computing gradients through the sampled execution trace.

---

**Algorithm 1:** Learning BOMs

**Input:** Probabilistic model $\mathcal{W}_\phi$, Prior, Observations $\tau$
**Result:** BOM with inferred parameters $\hat{\phi}$
**while** *not converged;*                                                                 /* Inference loop */
**do**

    $\mathcal{G} \leftarrow$ SAMPLE-STRUCTURE
    $\Theta \leftarrow$ SAMPLE-KINODYNAMICS
    $\hat{\tau} \leftarrow$ DIFFERENTIABLE-PHYSICS$(\mathcal{G}, \Theta)$
    $\tau_{\text{obs}} \leftarrow$ SAMPLE-OBSERVATIONS$(\hat{\tau})$
    loss $\leftarrow$ COMPUTE-LIKELIHOOD-OR-ELBO$(\tau_{\text{obs}}, \tau)$
    Update $\mathcal{W}_\phi$ ;                                                    /* Update BOM */
**end**

---

### 3.4 Gradient-based Bayesian inference of structure and scene parameters

Having specified our model $g_\phi$ and appropriate priors over the latent variables, we perform Bayesian inference to recover the posterior $p(\mathcal{G}, \mathcal{J}, \Theta | \tau_{\text{obs}})$ given the observed trajectories $\tau_{\text{obs}}$ (Eq 1).

A key benefit of using modern PPLs (such as Pyro in our case) is that nearly all of the effort is concentrated on model, prior, and likelihood specification. As such, inference is automated by a single function call, which enables us to experiment with a variety of inference schemes. We experiment with the following inference schemes, each of which presents trade offs between modeling assumptions, speed, and solution quality:

- **SVI** (Stochastic variational inference [3]): The posterior is approximated using a well-behaved distribution (called the variational distribution), allowing us to formulate a lower bound (ELBO), and optimize it by gradient descent.
- **HMC** (Hamiltonian Monte Carlo): A gradient-based MCMC scheme that uses Hamiltonian dyanmics to propose samples close following a target distribution.
- **NUTS** (No-U-Turn Sampler): A variant of HMC that adaptively tunes step sizes of the HMC sampler, reducing manual tuning effort.

---

[3]SVI is *stochastic* in that it computes an approximation to the ELBO using a few samples drawn from the generative model. Eliminating the stochasticity would entail computing the full ELBO, which is computationally intractable in most real-world problems of interest.

| | Approach | Oracle? | *OpenCabinetDoor* Success(%) | Graph MSE | Joint(%) | IdE | *OpenCabinetDrawer* Success (%) | Graph MSE | Joint(%) | IdE |
|---|---|---|---|---|---|---|---|---|---|---|
| | Disconnected | ✗ | - | 0.813 | - | - | - | 0.772 | - | - |
| | Clique | ✗ | - | 0.188 | - | - | - | 0.229 | - | - |
| Privileged | SBI (SNPE) | ✓ | 16.67 | - | 100 | 0.67 | 4.76 | - | 100 | 0.65 |
| | SBI (SNLE) | ✓ | 19.05 | - | 100 | 0.53 | 7.14 | - | 100 | 0.54 |
| | Diff. sim. | ✓ | 95.23 | - | 100 | 0.01 | 90.48 | - | 100 | 0.04 |
| | Diff. sim. (tail init) | ✓ | 9.52 | - | 100 | 0.93 | 0 | - | 100 | - |
| Ours | BOM-SVI (Ours) | ✗ | 100 | 0.024 | 91.08 | 0.12 | 100 | 0.019 | 94.33 | 0.09 |
| | BOM-HMC (Ours) | ✗ | 100 | 0.081 | 90.58 | 0.23 | 100 | 0.063 | 93.92 | 0.21 |
| | BOM-NUTS (Ours) | ✗ | 100 | 0.080 | 90.33 | 0.22 | 100 | 0.061 | 93.58 | 0.21 |

Table 1: Evaluating structure discovery and kinodynamic prediction capabilities of BOMs. Success(%) indicates task success. Graph MSE measures the mean-squared error between the predicted and true graphs. Joint(%) evaluates the fraction of joints that were correctly characterized. IdE is an 'identification error' metric that measures the absolute relative error in kinodynamics parameters. We see that BOMs achieve higher success rates compared to privileged strategies (SBI, Diff. sim.) that assume access to ground truth object models.

## 4 Experimental Results

We conduct experiements to investigate the following questions about BOMs.

1. How effective are Bayesian object models (BOMs) at probabilistically modeling the structure of objects with unknown kinodynamics?
2. How effective are BOMs at characterizing distributions over kinodynamic parameters?
3. How do BOMs help real robots interact with previously unseen objects?
4. Are the inferred BOMs useful in downstream tasks such as tracking object parts?

### 4.1 Evaluating the structure and kinodynamic prediction capabilities of BOMs

We first evaluate the quality of tree-structured distributions estimated by BOMs for a variety of inference techniques, and against various inference techniques and relational inference schemes representative of current art.

**Experiment Setup**: To learn a BOM, we first record a 2-5 second pointcloud sequence of an action (external force) being applied on an object of interest. Each pointcloud is passed to an object part segmentation network. The tracked parts are used to initialize prior distributions for sampling center of mass and axis parameters. For each object, we use 8-20 such interaction trajectories to *learn* the parameters of the BOM.

**Dataset**: We use 3D assets provided by PartNet-Mobility [58]; in particular the *Cabinet* class for results reported in this section. We also use the *Dishwasher* and *Microwave* classes for additional experiments. These assets have ground-truth 3D meshes and kinodynamic parameters. Further, to evaluate robotic interaction capabilities, we use the *OpenCabinetDoor* and *OpenCabinetDrawer* tasks from the ManiSkill [59] benchmark. While the tasks in Maniskill use 67 train and 20 test instances, we modify this setup and use 42 instances for testing, owing to the sample efficiency of BOM model optimization. We use the remaining (45) instances to compute statistics that inform our data-driven prior. In all, we experiment on 142 simulated object instances from categories commonly found in household environments.

**Approaches evaluated**: We consider three broad classes of approaches. *Blackbox* approaches include recent work in simulation-based inference [60, 61] implemented as part of the SBI toolbox [62]. As a strong baseline that assumes access to privileged information, we also use a differentiable simulator [63] equipped with an *oracle* physically-accurate kinematic and mesh model of the scene, albeit with unknown dynamics parameters. Our suite of inference engines for learning BOMs include SVI, HMC, and NUTS.

**Discussion**: Table 1 compares all the approaches on a number of metrics. *Graph MSE* evaluates the mean-squared error between the true adjacency matrix and the predicted adjacency distribution. *Joint(%)* computes the accuracy by comparing the estimated joint type with the greatest likelihood against ground-truth. *IdE* measures the relative identification error in physical parameters (for Bayesian approaches, we take the median estimate of each parameter). *Success(%)* measures

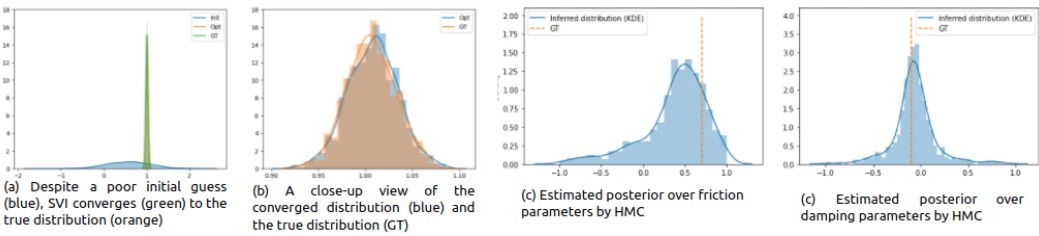

(a) Despite a poor initial guess (blue), SVI converges (green) to the true distribution (orange)

(b) A close-up view of the converged distribution (blue) and the true distribution (GT)

(c) Estimated posterior over friction parameters by HMC

(c) Estimated posterior over damping parameters by HMC

Figure 4: **Quality of physical parameters** estimated by BOMs.

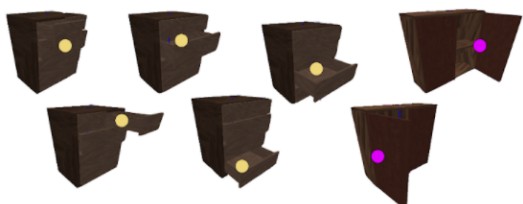

Figure 5: **Qualitative results**: Joint types plotted against the corresponding axis. Yellow indicates a prismatic joint, while magenta indicates a revolute joint.

the percentage of task success rates in ManiSkill.[4] Note that Diff. sim. (and variants), BOM-SVI, BOM-HMC, and BOM-NUTS all leverage gradients of the observation likelihood w.r.t. the parameters of the generative model. However, they have completely different objectives. While Diff. sim. variants attempt to find a local minimum of the observation (negative log) likelihood, BOM variants attempt to estimate a posterior distribution that can be used to draw samples. We use the median from a set of 10000 samples drawn from BOM-HMC and BOM-NUTS as the representative point for evaluation.

**BOMs infer accurate kinodynamic parameters.** We see from the *IdE* columns in Table 1 that BOMs infer physical parameters more accurately compared to gradient-free probabilistic inference methods (SBI-SNPE, SBI-SNLE). The only approach that outperforms BOMs on this metric is differentiable simulation – however, this approach uses privileged information in the form of ground-truth meshes, parts, and centers of mass. The primary source of error in BOMs is precisely because of errors induced in estimating centers of mass from pointcloud observations instead.

**BOMs accurately characterize distributions over dynamics parameters.** To evaluate the learned distribution of dynamics parameters in BOMs, we plot the inferred distributions over friction and damping parameters of a cabinet asset from PartNet in Fig. 4(c,d). Notice how the density of samples drawn from BOM-NUTS is concentrated near the true point estimate.

**BOMs are well-suited for robotic interaction.** From the *Success(%)* columns in Table 1, and the qualitative results in Fig. 5, we observe that BOMs accurately capture kinematic structure, which enables high success rates on robotic interaction tasks. We assume a known, deterministic policy is employed for each of the tasks. Object kinodynamic parameters in the (nondifferentiable) SAPIEN [64] simulator are initialized by sampling from the learned BOMs. Therefore, task success rates depend crucially on accurate joint type classification. For example, misclassifying a revolute joint of a cabinet door as a rigid joint instead will render the *OpenCabinetDoor* task impossible.

**BOMs are robust to initialization errors.** To investigate the robustness of BOMs and to assess the sensitivy of differentiable simulators (strongest-performing approach on the *IdE* metric, we devise a baseline "differentiable simulation (tail init)". In this approach, we wilfully set the initial guesses of a gradient-descent routine to be sampled from the tail of the true distribution. We see, at once, that the differentiable simulation approach is quite brittle – success rates drop by over 80%. BOMs, on the other hand, are robust to such initialization errors. Fig. 4(a,b) show the learned distribution over damping coefficients of a cabinet from PartNet. Notice how the initial guess distribution in

---

[4]Note that in our setting, task failures are primarily due to structure and/or kinodynamic identification errors and not due to the interaction policies themselves. Action selection (i.e., choosing the amount of force to be applied, and where to apply it) is a good avenue for future work.

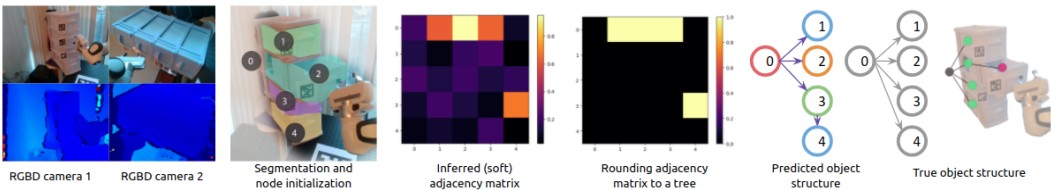

RGBD camera 1    RGBD camera 2    Segmentation and node initialization    Inferred (soft) adjacency matrix    Rounding adjacency matrix to a tree    Predicted object structure    True object structure

Figure 6: **Real-world BOMs** from a robot interacting with a cabinet to infer a distribution over its structure.

panel (a) has an extremely high-variance (a poor initial guess for variational inference), but upon optimization, the learned distribution is near-identical to the true underlying distribution.

## 4.2 Structure discovery in other physical systems

We also compare the ability of BOMs to recover structure against modern causal structure discovery methods – neural relational inference (NRI) [37] and a variant, factorised-NRI (f-NRI) [38]. We consider the spring-mass and charged particle system environments in [37]. Given a set of observed trajectories of particles

| Approach | Known structure | Learned structure |
|---|---|---|
| NRI [37] | 2.98e-8 | 5.2e-5 |
| f-NRI [38] | 2.33e-8 | 4.7e-5 |
| Ours | 2.21e-8 | 1.78e-6 |

Table 2: Structure discovery evaluation

(coupled by unobserved springs or charges), the aim is to recover the latent structure that best explains the trajectories. The quality of the recovered structure is evaluated by using the probabilistic model to predict future trajectories of each particle, and evaluating mean-squared error with respect to ground-truth. NRI [37] recovers a point estimate of the underlying scene structure. f-NRI [38] is an improved variant that extends NRI to model multiple interaction types. The 'known structure' setting only recovers physical parameters (masses, spring constants, charges), whereas the 'learned structure' setting recovers both structure and physical parameters. Notice how BOM-SVI perfroms significantly better than both these approaches.

## 4.3 Real-world experiments

We also evaluate the structure learning abilities of BOMs in a real scene. In this experiement, a robotic manipulator (Franka Emika Panda arm) interacts with a cabinet. Color and depth images are captured by a pair of Intel RealSense cameras. Part segmentation on the color images—currently performed manually for the first 5 frames (and subsequently tracked using the Lucas-Kanade algorithm)—initializes the node structure for a BOM. In Fig. 6, we visualize the experiment setup and the inferred scene structure. While the observed trajectory (a single interaction) is insufficient to explain the structure of all movable links in the object, we observe that the tree structure recovered quite accurately models the moving components of the object.

## 4.4 Deploying BOMs for object part tracking

We evaluate the applicability of learned BOMs in part tracking. Once trained, BOMs can be used to analyze interventions, predicting the outcomes of novel actions. Table to the right shows the performance of BOMs on tracking the parts of a cabinet by plotting the mean squared error of the locations of the center of masses.

| Approach | MSE (metres) |
|---|---|
| Const. vel. | 10.128 |
| Const. acc. | 37.025 |
| BOM-SVI | 0.1704 |

Table 3: Tracking object parts with BOMs

We compare this against two baseline approaches, using a constant velocity model and a constant acceleration model respectively (common choices in Bayes filters).

## 5 Discussion and Limitations

This work presented BOMs – an approach to build computational models of everyday objects while characterizing uncertainty over structural and kinodynamic parameters. BOMs are effective and sample-efficient; they can be built from even a single real-world interaction. However, BOMs currently assume static kinematic tree structures, and rely on generic inference schemes (SVI, HMC), and require 7-30 minutes of runtime per interaction sequence. We refer the interested readers to our supplemental material for more details.

**Acknowledgments**

KMJ acknowledges support from the NVIDIA fellowship.

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
