# OpenReview forum: "Bayesian Object Models for Robotic Interaction with Differentiable Probabilistic Programming"
_robot-learning.org/CoRL/2022/Conference — CoRL 2022 Poster_

### Official Review · Reviewer_b3K8 · 2022-07-28

**Originality:** Fair
**Technical Quality:** Good
**Clarity Of Presentation:** Fair
**Impact:** 2

**Recommendation:**

Weak Reject: I recommend rejecting the paper, but will not argue for my recommendation if the majority of other reviewers have a different opinion.

**Summary:**

In robotics applications, there are often multiple moving parts which may be connected in different ways,
for example, with rigid, prismatic or revolute joints.
Further to this, robots are often accompanied by forward kinodynamics models that can be used to simulate trajectories given
the scene structure (models of the moving parts) and an action sequence.
This paper proposes a method for inferring the scene structure as a graph (part types, joint types) and kinodynamic parameters given trajectories observed from the true system.
They present a method for approximating a likelihood function using a simulator.
As such, I believe that their method heavily leverages ideas from simulation-based inference, however,
this is not stated in the paper.
They also present a novel differentiable mechanism for sampling tree-based random variables which they exploit
for gradient-based approximate inference strategies.



**Issues:**

1. The likelihood in Equation 1 looks a little funny to me.
    - $p(\tau \mid \mathcal{G}, \mathcal{J}, \Theta)$ is the likelihood where $\tau$ represents the observed variable and the latent variables are $\mathcal{G}$, $\mathcal{J}$ and $\Theta$.
        - So what is $p(\tau \mid \tau_{\text{obs}})$? I don't think $p(\tau \mid \tau_{\text{obs}})$ makes sense. It looks like some form of predictive posterior with the latent variables integrated out.
    - Also, shouldn't there be a normalising constant if using equals sign?
        - I think it should be either $p(\tau \mid \mathcal{G}, \mathcal{J}, \Theta)= \frac{1}{Z}\prod_{t=0}^{T} \exp \left(-\beta_{t}\left\|\tau_{i}-\tau_{\text {obs }, i}\right\|^{2}\right)$ or $p(\tau \mid \mathcal{G}, \mathcal{J}, \Theta)\propto\prod_{t=0}^{T} \exp \left(-\beta_{t}\left\|\tau_{i}-\tau_{\text {obs }, i}\right\|^{2}\right)$
    - I am also not sure what the $i$ subscript represents? I assume it is meant to represent the time step and should be $t$?

2. I think that this work needs to be reformulated through the lens of simulation based inference.
    - Firstly, the relevant literature needs to be cited in the introduction.
    - Secondly, I found some of the language used throughout quite confusing, so I think reformulating this work as simulation based inference would make it much easier to follow.

3. I find Algorithm 1 quite misleading. Surely the algorithm changes depending on the inference scheme? For example, SVI computes the Evidence Lower BOund (ELBO), not the likelihood like Algorithm 1 suggests.

4. I also find details of the SVI implementation to be lacking.
    The success of SVI heavily relies on the variational approximation and the evidence lower bound.
    I think it is important to detail the variational posteriors that are being used along with a list of
    the parameters which are being optimised and which are fixed (if any).
    It would also be nice to see the objective (lower bound) that is being used as I am slightly unsure about using SVI
    in this setting.
    How are the KL divergence terms calculated, what are the local variables (factorised over data points) and what are the global variables?

5. Line 252: Is this a poor initial guess? It looks like a good initial guess to me! The mean is close to the true value and the high variance means that the true value has some probability mass, which usually allows variational inference to easily find a mode of the posterior close to the true value.

6. It would also be helpful to know how the inverse variance $\beta_{*}$ in the likelihood (Equation 1) is picked? Or is it learned?

Minor fixes:

- Algorithm 1 is not referenced?
- Figure 2 is not referenced?
- Line 157: $u_{l}j$ looks like a typo?
- Line 161: "This scheme provably samples a tree". Where is the proof? Add a reference.
- Line 200: Sentence doesn't make sense. Have you repeated the various inference techniques part?
- Line 201: should "current art" be "current state-of-the-art"?
- Line 204: Space before ".The"?
- Line 222: Should NLL be MSE?
- Line 230: What are SBI-SNPE and SBI-SNLE? Introduce acronyms.
- Figure 4: Increase legend and axis font size
- Figure 4c/4d are referenced before 4a/4b. Consider switching the paragraphs around or switching the order of sub figures.
- Figure 6: Increase font size of sub captions
- Figure 6: I can't read the axis or colorbar values in the middle two plots!
- Table 2: what do the numbers represent? Add units and more info in caption.

**Quality Of The Limitations Section:**

Limitations are addressed clearly

**Reviewer Expertise:**

4: The reviewer is confident but not absolutely certain that the evaluation is correct

**Robotics Focus:**

Sufficient demonstration on hardware

**Strengths And Weaknesses:**

**Strengths**

1. Given a simulator which can simulate trajectories given different combinations of scene structure and kinodynamic parameters, their method is able to infer a posterior distribution over the scene structure and kinodynamic parameters given trajectories observed from the true system.
    - In comparison to previous methods, which can update simulation parameters given observations from the true system, their method also infers structure. For example, given two objects that rotate relative to each other, their method would infer that they are likely attached via a revolute joint.
    - I think this is pretty neat.
2. This relies on defining tree-structured random variables and they propose a novel differentiable mechanism for sampling such tree-structured variables. As such, they can adopt gradient-based inference schemes.
    - Although many dynamics models are differentiable, they are often not differentiable with respect to an object's structure.

**Weaknesses**

1. The paper is not positioned well against the relevant literature on simulation-based inference which makes for a confusing read. This paper presents a simulation-based inference method but does not mention simulation-based inference until the experiments section, where they compare their method to two alternative simulation-based inference approaches.
    - I was initially very confused by their likelihood function in Equation 1 and what they refer to as their generative model
        $g_{\phi}(\tau \mid \mathcal{G}, \mathcal{J}, \Theta)$.
        In my eyes, this work is simulation-based inference, as they do not have access to a likelihood function that relates
        observed trajectories $\tau_{\text{obs}}$ to their latent variables ($\mathcal{G}$, $\mathcal{J}$ and $\Theta$).
        Instead, they have a simulator $g_{\phi}(\tau \mid \mathcal{G}, \mathcal{J}, \Theta)$ which depends on
        their latent variables ($\mathcal{G}$, $\mathcal{J}$ and $\Theta$).
        The goal is to obtain the posterior over the latent variables given the observed trajectories $p(\mathcal{G}, \mathcal{J}, \Theta \mid \tau_{\text{obs}})$. Usually this is calculated with Bayes rule as follows:
        \begin{equation}
        p(\mathcal{G}, \mathcal{J}, \Theta \mid \tau_{\text{obs}}) = \frac{p(\tau \mid \mathcal{G}, \mathcal{J}, \Theta) p(\mathcal{G}, \mathcal{J}, \Theta)}{\int\int\int p(\tau \mid \mathcal{G}, \mathcal{J}, \Theta)p(\mathcal{G}, \mathcal{J}, \Theta)\text{d}\mathcal{G} \text{d}\mathcal{J} \text{d}\Theta}
        \end{equation}
        However, they do not have access to a likelihood function $p(\tau \mid \mathcal{G}, \mathcal{J}, \Theta)$ and must
        resort to simulation-based inference.
    - If I understand correctly, $g_{\phi}$ simulates a trajectory $\tau$ for a sampled set of latent variables and then their "likelihood" is approximated using the exponential of the MSE, given by,
        \begin{equation}
        p(\tau \mid \mathcal{G}, \mathcal{J}, \Theta)\propto\prod_{t=0}^{T} \exp \left(-\beta_{t}\left\|\tau_{i}-\tau_{\text {obs }, i}\right\|^{2}\right) \quad \text{where} \quad \tau_0, \ldots, \tau_{T} \sim g_{\phi}(\tau \mid \mathcal{G}, \mathcal{J}, \Theta).
        \end{equation}
        I think this is a form of simulation-based inference known as approximate Bayesian computation (ABC). See Cranmer et al., 2020.
    - However, I could be wrong as I am not well versed in simulation-based inference.

**References**

Cranmer, Kyle, Johann Brehmer, and Gilles Louppe. "The frontier of simulation-based inference." Proceedings of the National Academy of Sciences 117.48 (2020): 30055-30062.

**Summary Of Recommendation:**

This paper presents a simulation-based inference (SBI) approach without explicitly stating their approach is SBI.
Not only does the paper lack a discussion of the relevant literature but at least one reviewer (me) has been assigned
this paper with little to no experience in SBI.
Unfortunately, I think this is due to an oversight by the authors which needs to be addressed before publication.
However, I do think the approach presented in this paper has the potential to be a good piece of work,
the authors just need to position their work against the relevant SBI literature and resubmit, stating clearly that
their work is SBI.

---

### Official Review · Reviewer_rwAv · 2022-07-31

**Originality:** Very Good
**Technical Quality:** Good
**Clarity Of Presentation:** Fair
**Impact:** 4

**Recommendation:**

Weak Accept: I recommend accepting the paper, but will not argue for my recommendation if the majority of other reviewers have a different opinion.

**Summary:**

This work presented a probabilistic object representation, called Bayesian object models (BOMs) for robotic interaction. This representation encodes the structural properties such as scene structure or part structure of articulated objects (e.g. a cabinet with multiple drawers), and the physical kinodynamics attributes necessary for interaction (e.g. joint category and joint parameters of cabinet drawers). Concretely, relying on probabilistic programming with a proposed tree structure sampler and differential physical engines, BOMs infer the posterior distribution over the aforementioned types of certain objects, i.e. scene structure, physical properties given the prior of them and the likelihood of the object trajectories during task execution. Its usefulness on scene structure discovery, physical parameter estimation and interactive robotic manipulation has been empirically demonstrated on different setups such as Maniskill data set, other physical systems and a real robotic arm.

**Issues:**

As listed in the section of Strengths And Weaknesses, the reviewer would suggest the authors to put higher priority on the clarify of the paper and then the methodological details.

**Quality Of The Limitations Section:**

Additional details required

**Reviewer Expertise:**

3: The reviewer is fairly confident that the evaluation is correct

**Robotics Focus:**

Sufficient demonstration on hardware

**Strengths And Weaknesses:**

Overall, the reviewer is highly in favor of the idea of building a rich computational mental model of objects for robotic interaction by utilizing the recent advances of probabilistic programming and differentiable physical engines. This work on a deeper combination of these two fields through integrating probabilistic programming into a robotic software stack is inspiring and beneficial for both communities on developing better theories and methodologies that are suited for wider scenarios in practice and able to addressed urgent concerns like robustness and trust of the system. In order to achieve this idea, the authors formulate the problem in a probabilistic manner, design the core elements within the pipeline such as the prior, likelihood as well as the inference engines and embed them in to a differential physics engines. Specifically, confronted with in-differentiability of tree sampling in generating scene structure, the authors overcome this by enforcing intuitive constraints (i.e. a opportunistic rounding scheme) to a differentiable DAG sampler. The extensiveness of empirical validation in terms of effectiveness on different aspects of the proposed model is also kindly appreciated by the reviewer. Though the real-world experiment is not convincing enough for a large credit, the reviewer considers this as a last-but-not-least complementary part to the experiment.

Nevertheless, the reviewer would still like to express several concerns regarding the idea and technical details, and provide suggestions for improving the paper and making it ready for publication.

### Presentation
Regarding the presentation clarify, the reviewer conveys the concern pressingly and tries to list the suggestions in the following.
- nearly all figures are in a super low resolution.
- it would be great to explain the term `generative model`, which means normally a model describing an unconditional random variable in contrast to the discriminative one for the conditional random variable. Because the main task of this work is to infer the posterior distribution over the scene structure and physical parameters of the task-specific objects, which the reviewer would not consider it as a generative model.
- Sec. 1: the concrete usefulness (e.g. which exact tasks to apply) is not clearly described.
- Sec. 3:
     - the definition of notations such as $theta$, $J$ is unclear and ambiguous, i.e. multiple names for one letter (e.g. $J$).
     - in the problem definition, it is so unexpected to see a sudden appearance of unseen terms such as "a joint type", "joint parameters" and so on in the text, the reviewer suggests the authors to make a distinctive **separation** between the description of a problem formulation and an example of the problem. It seems that these two parts are mixed in this sub-section.
     - Eq (2) is more clear to explain the problem, which should be moved into the problem definition part.
     - in sub-sec 3.1., there is a sentence "BOMs exacerbate...", should "exacerbate" be replaced by some words like "resolve"?

- Sec. 4.: The order of the problems raised at the beginning does not match with the organization of the following sub-sections, which confuses the reader a lot. Besides the mis-matching problem, the clarify of presenting the experimental results has large room to improve.

- There are some more minor typos in Sec4. please correct them.

### Methods
- Since there is no clear definition of the physical parameters ($J$ and $\theta$) and edge types ($e$) of the structure graph, the first concern is the generalibility of this idea. How general can it be applied to other tasks instead of just the cabinet drawer/dishwasher/microwave (no results for the latter two in the paper) opening?

- There is no clear definition of trajectories ($\tao$) used in the likelihood computation. Is it only the position or 6D pose of the point cloud? Is it possible to use another way to represent the evidence, such as segmentation of bounding box of the moving parts in 2D images? What's the motivation and benefits to use the trajectories of 3D point cloud while it might be more computational intensive?

- There is no clear definition of the physical parameters (i.e. the kinodynamics), though they are details. It would be great to describe them and improve the self-containness of the paper. And the reviewer has the feeling that they might be instructive on specifying a suitable prior and inference method.

- What differential physics engine is used? Is it possible to visualize them (e.g. the drawer opening task) for both the prior and posterior?

- What's the insight for choosing the inference methods in this contexts? The reviewer asks this question because HMC is generally considered better than SVI, which is not shown in the results. The second related question is that which approximate distribution is used in SVI, which matters a lot for the performance.

### Experiments
- The results in Table 1. makes the reviewer confused because the better Joint accuracy and $IdE$ of Diff. sim leads to a lower success rate BOMs. On the other hand, the authors explain the results with an argument that "misclassifying a revolute joint of a cabinet door as rigid joint instead will render the `OpenCabinetDoor` task impossible", but the BOMs can achieve 100% success rate with only ~90% Joint prediction accuracy. Please make this part more clear.

- Which question does sub-section 4.3 answer?

- The reviewer considers the claimed feature of `sample-efficient` as weak, because there is no comparable experimental result to show it.

### Limitation
- this part would be great if the overconfidence behavior of SVI could be explained in more details.


**Summary Of Recommendation:**

Even though there are several concerns on the clarify and method parts (as explained above), the novelty of the idea and the potential impact on promoting probabilistic programming in robotics persuade the reviewer to accept this work weakly.

---

### Official Review · Reviewer_8ymB · 2022-08-01

**Originality:** Good
**Technical Quality:** Very Good
**Clarity Of Presentation:** Very Good
**Impact:** 4

**Recommendation:**

Weak Accept: I recommend accepting the paper, but will not argue for my recommendation if the majority of other reviewers have a different opinion.

**Summary:**

In this article, the authors present Bayesian Object Models (BOMs) which are generative probabilistic models that encode both the structural and kinodynamic attributes of an object. It is formulated as a Bayesian inference problem over a generative model, which is implemented in a Probabilistic programming languages, given a number of real-world interactions. At the end, the framework allows to learn a distribution over both the scene structure and its physical parameters. Several simulations and experiments who the benefit of the proposed approach.

**Issues:**

major:
- see above

minor:
- Fig 3: "r triangular adjacency mat" matrix?

**Quality Of The Limitations Section:**

Limitations are not well addressed

**Reviewer Expertise:**

3: The reviewer is fairly confident that the evaluation is correct

**Robotics Focus:**

Sufficient demonstration on hardware

**Strengths And Weaknesses:**

Strength:
- In general, the paper is well written but some more details would improve it
- Strong experiments with comparison to state-of-art are techniques
- The robustness against initialization errors of BOM are very beneficial in practice I guess

Weaknesses:
- P5, 158; "we find the first entry ulj in the soft upper-triangular matrix U above a threshold δ and round it up to 1" Why are you using the first element here? It seems like there might be better ways to achieve a tree.
- The "limitation section" discusses the general limitations of inference approaches rather than limitations of the actual proposed method
- The presented approach seems to be computational expensive but there is no evaluation/comparison on this.

**Summary Of Recommendation:**

The paper is well written and presents an interesting idea with potential impact on the community. However, the limitations of the proposed approach are rarely addressed.

---

### Official Review · Reviewer_JsgS · 2022-08-04

**Originality:** Good
**Technical Quality:** Good
**Clarity Of Presentation:** Fair
**Impact:** 3

**Recommendation:**

Weak Reject: I recommend rejecting the paper, but will not argue for my recommendation if the majority of other reviewers have a different opinion.

**Summary:**

This paper presents a tree-based probabilistic model for estimating the kinodynamic properties of a scene. The tree's nodes are the objects, their links, etc, and edges are properties like joint type, mass, friction, damping, and 6D transformations. Authors propose a differentiable generative model for sampling trees given some prior distribution. Next, the parameters of the prior distributions are updated to match observed trajectories. The proposed approach can estimate the kinodynamic properties of interactive scenes like cabinets, dishwashers, etc. Comparative experiments against more privileged methods like differentiable simulators show that the proposed approach is more robust to poor initializations.

**Issues:**

1. Experimental analysis that explains the performance of the proposed approach over baselines.
2. Discussion of limitations like variable, unknown number of trees in the scene
3. Better motivation for the chosen problem. When can we not assume that we know simple kinematic things like joint types, and why not just infer dynamic properties of the scene?

**Quality Of The Limitations Section:**

Limitations are not well addressed

**Reviewer Expertise:**

3: The reviewer is fairly confident that the evaluation is correct

**Robotics Focus:**

Sufficient demonstration on hardware

**Strengths And Weaknesses:**

Strengths:
1. An interesting, sensible approach for estimating properties of an unknown scene
2. Competitive baselines, and good simulation experiments that show the strengths of the work
3. Real-world experiments that show the applicability to real robots

Weaknesses:
1. Unclear motivation of the problem: I am unclear on why can't we assume that we know the joint types in the scene? Simple categories like prismatic versus revolute are known about most object types, and can be estimated (even without interaction). Overall, I found the basic premise of the paper somewhat ill-motivated.
2. Missing analysis of experimental results. Authors show that proposed approach can find good solutions even when initialized from a bad prior, while diff sim cannot. Why? Isn't the proposed approach also gradient based? So why is it more robust to poor initialization?
3. Discussion of limitations: How does this approach deal with situations where there might be multiple trees (robot is not in contact with drawers)? And what happens when the number of trees in the scene change (robot makes contact, and the trees combine)? I think assuming just one tree is kind of limiting for most robotic scenes, so this should be discussed in the paper.

**Summary Of Recommendation:**

I think this is an interesting idea, and the paper presents nice experiments. However, I find experimental analysis lacking, and I don't have a good intuition for why the proposed approach does better than the comparisons. Additionally I found the motivation and limitations of the work lacking.

---

> ### Comment · Reviewer_JsgS · 2022-08-28
> **response to authors**
>
> Thanks for your response. I appreciate the additional motivation, and discussion. My question was not about multiple objects, but instead about variable number of graphs. Thanks for adding this to the limitations of the work.
>
> I would have liked to see a more detailed analysis and response regarding the differences between DiffSim and BOM. For example, how does the posterior estimate of BOM differ (which components are different, why do they matter?). As it stands, I am still unclear on what is different between current work, and what exactly makes the performance better. As a result, I am sorry but I still will keep my rating the same as before (weak reject).

---

### Meta-Review · Area_Chair_67fg · 2022-09-07

**Recommendation:** Accept (Poster)
**Confidence:** 3

**Metareview:**

The authors propose a simulation-based inference framework that learns the structure (objects in the scene) and the physical kinodynamics properties of the objects. Through sampling robot manipulation trajectories can be inferred.

Strength:

    -Relevant and interesting approach that considers uncertainties
    -Probabilistic generative model that can be used for inference tasks
    -Extensive baseline comparison
    -Extensive evaluation in simulation and real robot experiments

Weakness:

    -Assumptions and limitations not adequately discussed (also to motivate the approach)
    -Related work on simulation-based inference not discussed in sufficient detail

Update: Modified manuscript with detailed discussions on SBI and limitations.


**Best Paper Nomination:**

No